# Chemical and Physical Ionic Liquids in CO_2_ Capture System Using Membrane Vacuum Regeneration

**DOI:** 10.3390/membranes12080785

**Published:** 2022-08-15

**Authors:** José Manuel Vadillo, Guillermo Díaz-Sainz, Lucía Gómez-Coma, Aurora Garea, Angel Irabien

**Affiliations:** Departamento de Ingenierías Química y Biomolecular, Universidad de Cantabria, ETSIIT, 39005 Santander, Spain

**Keywords:** carbon dioxide capture, membrane vacuum regeneration, hollow fiber membrane contactor, chemical IL [emim][Ac], physical IL [emim][MS]

## Abstract

Carbon Capture Utilization and Storage technologies are essential mitigation options to reach net-zero CO_2_ emissions. However, this challenge requires the development of sustainable and economic separation technologies. This work presents a novel CO_2_ capture technology strategy based on non-dispersive CO_2_ absorption and membrane vacuum regeneration (MVR) technology, and employs two imidazolium ionic liquids (ILs), [emim][Ac] and [emim][MS], with different behavior to absorb CO_2_. Continuous absorption–desorption experiments were carried out using polypropylene hollow fiber membrane contactors. The results show the highest desorption behavior in the case of [emim][Ac], with a MVR performance efficiency of 92% at 313 K and vacuum pressure of 0.04 bar. On the other hand, the IL [emim][MS] reached an efficiency of 83% under the same conditions. The MVR technology could increase the overall CO_2_ capture performance by up to 61% for [emim][Ac] and 21% for [emim][MS], which represents an increase of 26% and 9%, respectively. Moreover, adding 30%vol. demonstrates that the process was only favorable by using the physical IL. The results presented here indicate the interest in membrane vacuum regeneration technology based on chemical ILs, but further techno-economic evaluation is needed to ensure the competitiveness of this novel CO_2_ desorption approach for large-scale application.

## 1. Introduction

Great efforts to reduce carbon dioxide (CO_2_) emissions from the industrial and energy sectors (decarbonization) are crucial to reach the commitment to have net-zero greenhouse gas emissions by 2050 [1]. In this context, post-combustion Carbon Capture, Utilization, and Sequestration (CCUS) technologies are currently gaining interest due to their potential to significantly capture CO_2_ from large emission sources, including, on the one hand, thermal and power generation plants, which contribute greatly to the increase of CO_2_ in the atmosphere, and on the other hand, non-energy industrial sectors, such as the cement, chemical, and steel industries, where there is currently no real alternative to reach net-zero CO_2_. The different CCUS approaches, depending on the application of the captured carbon dioxide technology, involve the capture of CO_2_ from the output gas of industrial processes for permanent storage in geological cavities (CCS) [2] or use as a resource for carbon-based products (CCU) [3].

Solvent-based absorption–desorption technology, which typically separates CO_2_ from flue gas in packing columns, is presented as one of the most mature technologies. Nevertheless, the main challenge is to achieve a reduction in energy consumption for CO_2_–rich solvent regeneration carried out in the desorption column, which is estimated to constitute 80% of the total carbon capture system energy required [4]. Focusing on this, Membrane Vacuum Regeneration (MVR) technology has been proposed as a promising CO_2_ desorption process due to its potential for reducing the energy needed for regeneration with respect to conventional packed columns [5,6,7,8]. Using the MVR system, CO_2_ is desorbed in a hollow fiber membrane contactor (HFMC) from the rich solution by vacuum. The application of vacuum for CO_2_ desorption decreases the solvent regeneration temperature required and, therefore, the total energy needed for the CO_2_ capture system [9]. In addition, the lower operating temperatures of MVR technology increase the applicability of polymeric hollow fiber membrane contactors (HFMC), which have advantages, such as low production cost, hydrophobicity, commercial availability, and a wide range of chemical and morphological tunability—but are not suitable for high operating temperatures [4,10].

Until now, the most widely-used solvents in the CO_2_ desorption process with MVR technology are amine-based solutions, mainly due to the low cost, low viscosity, and high CO_2_ absorption capacity—even at low CO_2_ partial pressures [8]. In this context, Kosaraju et al. [11] demonstrated the feasibility of CO_2_ membrane stripping using commercial PP membrane contactors through long-term running (55-day test). Fang et al. [12], Yan et al. [13], and Wang et al. [14] screened 23 types of alkanolamines for MVR and experimentally evaluated the relationship between solvent composition and MVR efficiency. Listiyana et al. [6] conducted experiments on CO_2_ desorption in PP HFMC using activated amines to increase the CO_2_ regeneration efficiency and reduce the cost of the solvent. By indicating the focus on energy saving, Nii et al. [15] showed that MVR technology could effectively employ low-temperature energy or waste heat in power plants. However, several drawbacks have been reported for the amine-based CO_2_ desorption process, such as an energy intensive regeneration requirement, high absorbent loss, degradability, and corrosiveness of the HFMC, which promote the research of alternative solvents with better properties to address these disadvantages [16].

In this sense, ionic liquids (ILs) are presented as potential alternatives due to their special features for carbon capture, such as their high CO_2_ uptake capacity, negligible vapor pressure, wide operation liquid temperature range, and tunability [17,18]. ILs are divided into two main categories: non-functionalized room temperature (RTILs) and task specific (TSILs). The main difference between these two types of sorbents is that while RTILs behave like common physical absorbents for gases represented by Henry’s Law constant, TSILs present both physical and chemical absorption and, consequently, may absorb more CO_2_. However, the solvent regeneration process using TSILs is very energy-intensive due to chemical bonding [19].

Recent trends focused on CO_2_ desorption by coupled MVR technology using ILs may be resumed in the efforts of: (i) studying IL-based membrane contactors focused on solvent–membrane compatibility (Mulukutla et al. [20] and Bazhenov et al. [21]), and (ii) covering the design, modeling, and simulation for low-temperature CO_2_ desorption using different ILs to address the influence of operating variables (Lu et al. [22], Simons et al. [23] and Vadillo et al. [10]). Therefore, it is necessary to investigate the influence of IL nature (physical or chemical absorption) in the solvent regeneration performance not only for thermodynamic and kinetic IL properties (e.g., viscosity, CO_2_ solubility), but also with CO_2_ desorption process simulations. In this context, experimental data about the CO_2_ desorption process with MVR technology and ILs with both physical and chemical nature could be helpful to identify the key properties of solvents, considering the extra degree of freedom in ILs design provided by the tunability property [24].

In this work, the ILs, 1-Ethyl-3-methylimidazolium acetate ([emim][Ac]) and 1-Ethyl-3-methylimidazolium methyl sulfate ([emim][MS]), have been chosen as promising candidates as chemical and physical ILs, respectively. The effect of temperature and vacuum level were analyzed on the desorption performance and CO_2_ desorbed flux. Finally, the overall absorption–desorption system performance was calculated and discussed for both ILs at different operational conditions of solvent temperature, vacuum level, and the addition of water in the ILs. This evaluation would help the reader to recognize different types of ILs, which provides a blueprint for solvent selection in the field of the non-dispersive absorption–desorption process using membrane contactors and vacuum desorption as a promising carbon capture technology.

## 2. Materials and Methods

### 2.1. Materials and Characterization

The feed gas of the CO_2_ capture system was composed of 15% carbon dioxide (99.7, Air Liquide^TM^, Madrid, Spain and 85% nitrogen (99.9%, Air Liquide^TM^, Madrid, Spain), which was in the range of typical electro-intensive industries (10–25%) [25].

The IL [emim][MS] (≥95%) was supplied by Sigma Aldrich^TM^ and was selected due to its high values of surface tension and contact angle, moderate values of viscosity, and the presence of physical absorption, which potentially decreases the energy consumption during the solvent regeneration process [26]. The process performance was compared with previous works using the IL [emim][Ac] (≥90%) provided by Sigma Aldrich^TM^ Darmstadt, Germany; which presents chemical absorption (also called chemisorption) [10]. Table 1 shows the identification of the studied solvents.

To ensure the stability of the ILs for further CO_2_ capture experiments, the ILs’ decomposition temperatures were calculated by thermogravimetric analysis using a TGA-60H Shimazdu Thermobalance (Izasa, Japan). The viscosity was measured using a rotational viscometer at room temperature. In contrast, in order to evaluate the hydrophobicity/hydrophilicity and wetting behavior of the membranes, the static contact angles between the membrane and different ILs were estimated by the sessile liquid drop method using the contact angle quantification system (DSA25, Krüss, Hamburg, Germany). The contact angles were calculated at room temperature and atmospheric pressure. Adjustment of the picolitre dispenser (0.5 mm syringe) and camera image were also done before each component measurement. Then, a 2.0 µL drop with the desired component was deposited on the membrane’s surface at various sites (at least 5 points). Each value was obtained using the software provided through image recognition, and the average contact angle value was then considered.

### 2.2. Membrane Contactor

A hydrophobic polypropylene HFMC (1 × 5.5 MiniModule^TM^) supplied by Liqui-Cel^TM^ (3M Madrid, Spain) was used in parallel flow for the continuous absorption–desorption process. The module consists of mesoporous polypropylene hollow fibers with 40% porosity and a mean pore diameter of 0.04 μm. Figure 1 describes the flow configuration and Table 2 shows the specifications of the commercial membrane module used in this work.

### 2.3. Experimental Set-Up

In this section, the process units used for the CO_2_ capture experimental plant based on non-dispersive absorption using ionic liquids and MVR technology was described. Figure 2 shows a picture of the experimental system, consisting of:(1)Two hollow fiber membrane contactors that can operate interconnected, for both the non-dispersive absorption of the CO_2_ from the feed gas and the CO_2_ desorption by applying vacuum.(2)Two mass flow controllers (Alicat Scientific^TM^, Duiven, The Netherlands MC-gas mass flow controller Tucson, AZ, USA) to control the flows coming from the pure gas cylinders (CO_2_ and N_2_) to set the concentration and flow of the feed gas.(3)A digital gear pump (Cole-Parmer Gear Pump System^TM^ Vernon Hills, IL, USA, Mount Prospect, Vernon Hills, IL, USA, Benchtop Digital Drive, 0.017 mL·rev^−^^1^, 220 VAC, Saint Louis, MO, USA) to drive and maintain constant liquid flow during the continuous absorption–desorption process.(4)A closed vessel of tempered borosilicate glass (Pyrex^TM^, Paris, France) to contain and keep constant the temperature of the IL by means of a heater-stirrer.(5)Two gas analyzers (Geotech^TM^, G110 0-100%, Suffolk, UK) to measure the mass flow rate and CO_2_ concentration of the gas streams (feed gas, clean gas, and desorbed CO_2_ output). The analyzer is based on non-dispersive infrared spectroscopy (NDIR). The CO_2_ concentration in the output gas stream was monitored using the NGA Win-Control software.(6)A vacuum pump, with condenser included (Vacuubrand^TM^, PC 3001 VARIO PRO, Wertheim, Germany), to set the gas phase of the membrane contactor (used for CO_2_ desorption) at the desired vacuum pressure.

In addition, to visualize the system more clearly, Figure 3 shows the flow diagram of the continuous CO_2_ non-dispersive absorption–desorption plant using membrane contactors with the HFMC configuration.

The IL was recirculated through the lumen side of both HFMCs (absorber and desorber) in a closed loop. The solvent flow rate was kept constant by a digital gear pump. A heater was used in order to provide isothermal conditions during the continuous absorption–desorption process. The feed gas mixture was introduced through a counter-current at nearly atmospheric pressure across the HFMC absorber shell side in open-loop conditions with a constant flow rate, while the IL passes through the lumen side of the module absorbing the CO_2_. The CO_2_-rich IL was pumped into the HFMC desorber lumen side where the CO_2_ was transferred through the gas-filled membrane pores because of the shell side’s reduced pressure that was generated by the vacuum pump. The experiments of the continuous absorption–desorption process were running until the CO_2_ concentration on both gas outputs (clean gas and CO_2_ desorbed) were constant. Moreover, the experiments were carried out three times in order to ensure the results’ reproducibility. The data presented in the manuscript were the average values for the set of three experiments, with an experimental error within ±5%. Table 3 shows the process operating conditions.

### 2.4. Data Analysis

To analyze the performance of the CO_2_ desorption process based on MVR technology and its effect on the overall CO_2_ capture process, three parameters were mainly studied throughout the work.

(1)The CO_2_ desorption efficiency is calculated by Equation (1), where α_rich_ and α_lean_ are the CO_2_ loading in the IL (mol_CO2·mol_IL_^−1^) before and after one pass of IL through the HFMC desorber, respectively.


(1)
$$\mathrm{Desorption}\;\mathrm{eff}.\left(\%\right)=\frac{\alpha_{\mathrm{rich}}-\alpha_{\mathrm{lean}}}{\alpha_{\mathrm{rich}}}\times 100$$


(2)The overall CO_2_ capture efficiency, which is defined as the concentration difference in the HFMC absorber between the feed gas and the clean gas, is obtained by Equation (2), where C_(CO2,g)_^in^ (mol CO_2_·L^−1^ gas) is the CO_2_ concentration in the feed gas and C_(CO2,g)_^out^ (mol CO_2_·L^−1^ gas) is the CO_2_ concentration at the outlet of the module. The overall CO_2_ capture efficiency is important in order to study the influence of the MVR technology on the continuous absorption–desorption process.


(2)
$${\mathrm{Overall}\;\mathrm{CO}}_{2}\;\mathrm{capture}\;\mathrm{eff}.\;\left(\%\right)=\left(1-\frac{C_{\mathrm{CO}2,g}^{\mathrm{out}}}{C_{\mathrm{CO}2,g}^{\mathrm{in}}}\right)\times 100$$


(3)The CO_2_ desorbed flux (G_V_, mol·h^−1^ m^−2^) is estimated by Equation (3), where F_V_ is the CO_2_ flow rate desorbed from the HFMC desorber measured on the vacuum pump output (L·h^−1^), v_m_ is the molar volume of CO_2_ in ideal gas conditions (L·mol_CO2-1_), and A is the specific membrane area (m^2^).


(3)
$${\;G}_{V}=\frac{F_{V}}{v_{m}\;A}$$


## 3. Results and Discussion

The imidazolium ILs, [emim][Ac] and [emim][MS], were studied in this work as chemical and physical CO_2_ absorbents, respectively, for non-dispersive absorption technology using ionic liquid and MVR. In this section, the CO_2_ absorption capacity for each IL was discussed according to the equilibrium isotherms reported in the literature. Moreover, the experimental results of the absorption–desorption process operating at different MVR operating conditions (desorption vacuum pressures and solvent temperatures) were analyzed.

### 3.1. Absorption Properties

The equilibrium isotherms of the [emim][Ac]-CO_2_ and [emim][MS]-CO_2_ interactions were experimentally studied by Shifflet et al. [27] and Yim et al. [28]. Figure 4 shows the P–X diagram of CO_2_ solubility at different temperatures in both ILs used in this work.

IL [emim][MS] is considered a suitable absorbent for this system at higher partial pressures based not only on its good solvent properties (low volatility and very low viscosity), but also based on its behavior as a physical absorbent of CO_2_. IL [emim][Ac] is a better overall absorbent when attending to its isotherms and is specifically better at low CO_2_ partial pressures (where post-combustion CO_2_ capture processes work) due to the more thermodynamically favorable chemical reaction. Although the chemical IL, [emim][Ac], seems to be a better solvent for CO_2_ capture applications, the continuous absorption–desorption process not only depends on the CO_2_ absorption capacity of the solvent, but also on the MVR operation conditions (mainly vacuum pressure and liquid temperature), the CO_2_-IL chemical and/or physical interactions, and the contactor characteristics, such as membrane geometry and fluid dynamics [29]. In order to investigate the regeneration operation conditions and the IL nature influence on the CO_2_ desorption performance, a parametric study was considered in the next section.

### 3.2. ILs Characterization

Firstly, TGA analysis shows that the ILs remain without reducing more than 5% of their weight up to temperatures of 450 K and 470 K for [emim][MS] and [emim][Ac], respectively, as depicted in Figure 5. These issues confirm the ionic liquids’ stability and their capacity to carry out the absorption–desorption processes under real conditions without damage and degradation.

Moreover, the viscosity of both ILs was measured to ensure the good quality and purity of the samples. In addition, these values were compared with the previous data reported in the literature [30,31]. Finally, the contact angle demonstrates the hydrophilic or hydrophobic characteristics of the membranes with the ILs. Hydrophilic surfaces show low water contact angle (<90°) and hydrophobic surfaces show high water contact angle values. Table 4 summarizes the viscosity and measured contact angle values of both the [emim][Ac] and [emim][MS] ILs. Thus, it is possible to say that the polypropylene hollow fiber membranes used in this work present a hydrophobic character with the ionic liquids. Therefore, membrane wetting is highly avoided.

### 3.3. Parametric Study of Desorption Process

The following part of this study discusses the experimental results (CO_2_ desorption performance and CO_2_ desorbed flux) with both the chemical and physical ILs at different operating conditions of the vacuum regeneration system. The process efficiency and the CO_2_ desorbed flux results, which vary with the operating vacuum pressure and liquid temperature, were analyzed in Figure 6 and Figure 7, respectively.

On the one hand, a lower CO_2_ partial pressure (higher vacuum applied) on the permeate side of the HFMC desorber promotes the CO_2_ mass transfer driving force through the membrane, which increases the desorption performance as a result of the higher CO_2_ desorbed flux. On the other hand, the CO_2_ desorption process efficiency and the CO_2_ desorbed flux increase with a higher solution temperature. This behavior in both ILs could be explained by the lower viscosity (µ) at higher temperatures, which increases the diffusivity of CO_2_ in the absorbent since the mass transfer coefficient is controlled by the liquid-phase mass transfer resistance. Moreover, the CO_2_ partial pressure increases because of the higher concentration gradient at higher temperatures, leading to an increase of the CO_2_ desorbed flux.

From these experiments, it is clear that the vacuum level and liquid temperature should be as high as possible. However, two process limitations have to be taken into account in order to avoid HFMC operational problems: (i) the pressure applied by the vacuum pump in the permeate side was recommended to be greater than 0.035 bar in order to avoid wetting phenomena and (ii) temperatures higher than 310–320 K may require more resistant membrane materials due to thermal and chemical constraints of the commercial polypropylene HFMC used in our work. Physical IL [emim][MS] showed lower CO_2_ desorption performance and CO_2_ desorbed flux at the experimentally-tested temperatures (289-313 K) than the corresponding IL [emim][Ac]. This could be explained due to the lower CO_2_ loading capacity of the physical IL that leads to a lower CO_2_ driving force through the membrane.

Furthermore, only chemical [emim][Ac] was able to reach the target of 90% desorption efficiency for vacuum regeneration using the commercial PP-HFMC at the operating conditions (module characteristics given in Table 1). Two main points, which are related to the vacuum pressure effect, could be concluded from the results shown in Figure 6 and Figure 7: (i) Chemical IL [emim][Ac] was less sensitive to vacuum pressure conditions, which could be explained by the fact that the chemical IL requires more energy to break the CO_2_–IL chemical bond, and (ii) physical IL [emim][MS] required the application of higher vacuum than the corresponding IL, [emim][Ac], to reach the same desorption efficiency, which increased the energy consumption for both the vacuum pump and compressor unit operation, resulting in a higher overall cost of the CO_2_ vacuum desorption process [32].

In summary, the chemical IL, [emim][Ac], shows a better CO_2_ desorption performance and higher CO_2_ desorbed flux at different vacuum pressures and liquid temperatures. This could mainly be explained due to the larger capacity of CO_2_ to be absorbed into chemical IL by chemisorption, which increases the CO_2_ driving force through the membrane in the CO_2_ vacuum desorption process. However, the total energy consumption of the solvent regeneration with chemical ILs, such as [emim][Ac], may be expected to be higher than physical ILs, such as [emim][MS], due to the extra energy required to reverse the CO_2_–IL chemical reaction as previously reported [33,34].

### 3.4. ILs Comparison in the Overall CO_2_ Capture

The desorption process efficiency and the CO_2_ desorbed flux were evaluated in the previous section at different operation conditions in a continuous steady-state absorption–desorption CO_2_ capture system by using chemical IL [emim][Ac] and physical IL [emim][MS]. In this section, the influence of the CO_2_ desorption stage (based on MVR technology) in the overall CO_2_ capture system was evaluated by Equation (3), which was described in Section 2.4 (Data Analysis). The membrane contactor specifications and the operational conditions were described in Table 2 and Table 3.

Figure 8 shows the experimental results of the steady state absorption–desorption system at two representative operating conditions in order to evaluate the influence of both solvent temperature (289 K and 313 K) and CO_2_ desorption vacuum pressure (0.2 bar and 0.04 bar) on the overall CO_2_ capture efficiency for each IL analyzed in this work.

For chemical IL [emim][Ac], higher solvent temperatures and CO_2_ desorption vacuum pressures enhanced the overall CO_2_ capture efficiency from 29% to 61%. Moreover, the temperature contribution on the CO_2_ capture performance was more significant than the CO_2_ desorption vacuum pressure (Pv) since the lower viscosity of the [emim][Ac] at higher temperatures and the chemisorption effect increases the CO_2_ absorption capacity. For physical IL [emim][MS], higher vacuum applied in the MVR process increased the overall CO_2_ capture efficiency from 12% to 21%. However, at the same operating vacuum pressure, the solvent temperature effect on the overall CO_2_ capture performance was not significant. This could be explained because of both: (i) the decrease in viscosity at the studied temperatures (289 and 313 K, respectively) was less considerable in the CO_2_ absorption capacity and (ii) the absence of CO_2_–IL chemical interactions.

Previous studies of the single absorption process (without desorption technology) have reported efficiencies of 20–35% with chemical IL [emim][Ac] [35] and 10–12% with physical IL [emim][MS] [31]. Furthermore, in this work, the non-dispersive absorption using ionic liquids and vacuum regeneration, at the most favorable operating conditions studied (0.04 bar and 313 K), increase the CO_2_ capture performance up to 61% and 21%, respectively, which represents a percentage increase compared with the only-absorption process of 26% for [emim][Ac] and 9% for [emim][MS].

Although better absorption–desorption performance is obtained operating at the highest possible temperature due to the lower viscosity of the ILs, the operating temperature was kept below 315 K according to the specifications of the commercial membrane module used in order to avoid chemical or thermal degradation. Therefore, the option of using aqueous [emim][Ac] and [emim][MS] was checked to decrease the solvent viscosities, thereby keeping the temperature in the operational range.

### 3.5. Influence of IL Water Content in the Process Performance

In this section, experiments were carried out to investigate the effect on the continuous steady-state absorption–desorption process of 30%-vol. water content in the ILs. The overall CO_2_ capture efficiency was evaluated in Figure 9 at a liquid temperature of 313 K and different vacuum pressures (0.2 and 0.04 bar) by Equation (3) described in Section 2.4 (Data Analysis).

The results indicate that pure IL [emim][Ac] and aqueous IL [emim][MS] were the most favorable for the overall absorption–desorption process performance. Aqueous [emim][MS] physically absorbed CO_2_ and the CO_2_ diffusion in the IL was dominant mainly due to the viscosity. In this context, the reduction of viscosity provided by the addition of water to the solvent resulted in an increase of 10% in the values of overall CO_2_ capture efficiency. Pure IL [emim][MS] has a higher viscosity, which increases the CO_2_ mass transfer resistance in the liquid side and results in a lower CO_2_ diffusivity [34]. However, unlike IL [emim][MS], IL [emim][Ac] chemically absorbed CO_2_ and the reaction rate was dominant in the CO_2_ absorption capacity compared to the effect of water addition. Furthermore, pure IL [emim][Ac] may provide a higher reaction rate and thus decrease the mass transfer resistance to the liquid side.

## 4. Conclusions

The main objective has been to study the behavior of ILs with different natures as CO_2_ absorbents in terms of process performance. The chemical IL, [emim][Ac], and the physical, [emim][MS], were chosen as representative ILs for this study.

The influence on the CO_2_ desorption efficiency and CO_2_ desorbed fluxes for different operating conditions has been studied for both ILs. In general, higher temperature and vacuum applied are beneficial to the overall process performance and CO_2_ desorbed flux. In particular, the chemical IL, [emim][Ac], showed better CO_2_ desorption performance and higher CO_2_ desorbed flux than physical IL [emim][MS] at the operating conditions studied. The maximum CO_2_ desorption efficiencies obtained in this work were 92% and 83% for chemical IL [emim][Ac] and physical IL [emim][MS], respectively. Both performances were at 0.04 bar vacuum pressure, 313 K temperature, and a 60 mL·min^−1^ liquid flow rate.

Moreover, the overall CO_2_ capture efficiency was evaluated in order to analyze the influence of vacuum regeneration on the continuous absorption–desorption process at different liquid temperatures and vacuum pressures. Higher CO_2_ capture efficiency was reached for both ILs at the lowest desorption pressure (0.04 bar) and the highest liquid temperature (313 K) studied in this work. The CO_2_ capture efficiencies were 61% and 21% for chemical IL [emim][Ac] and physical IL [emim][MS], respectively. Moreover, the addition of water to reduce the ILs viscosity was evaluated in terms of CO_2_ absorption–desorption system performance. The results indicated that aqueous physical IL [emim][MS] increases the CO_2_ capture while aqueous chemical IL [emim][Ac] decreases the process performance due to the loss of chemical reaction potential by adding water. However, the study of the counterbalance effect of water content into ILs requires further study since water content could significantly affect the CO_2_ mass transfer through the membrane, as concluded in this work.

As a whole, the process performances of the chemical IL seem to be better than that of the physical IL in the continuous absorption–desorption CO_2_ capture system, while physical ILs could be considered as promising energy-saving absorbents for CO_2_ capture by designing more advanced physical ILs capable of absorbing more CO_2_.

From the viewpoint of scale-up, coupled membrane contactors, IL-based processes were addressed as a process intensification for CO_2_ capture. However, more studies of the continuous absorption–desorption systems are needed to drive the industrial implementation and commercial viability of non-dispersive absorption technology using ionic liquids and vacuum regeneration.

## Figures and Tables

**Figure 1 membranes-12-00785-f001:**
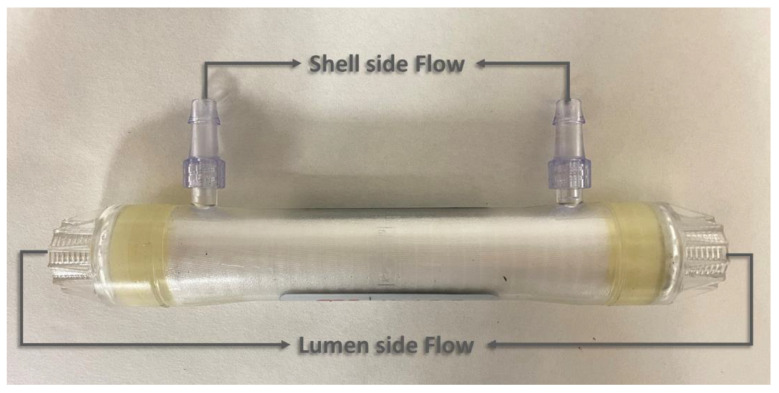
Commercial polypropylene hollow fiber membrane contactor in parallel configuration (1 × 5.5 MiniModule^TM^).

**Figure 2 membranes-12-00785-f002:**
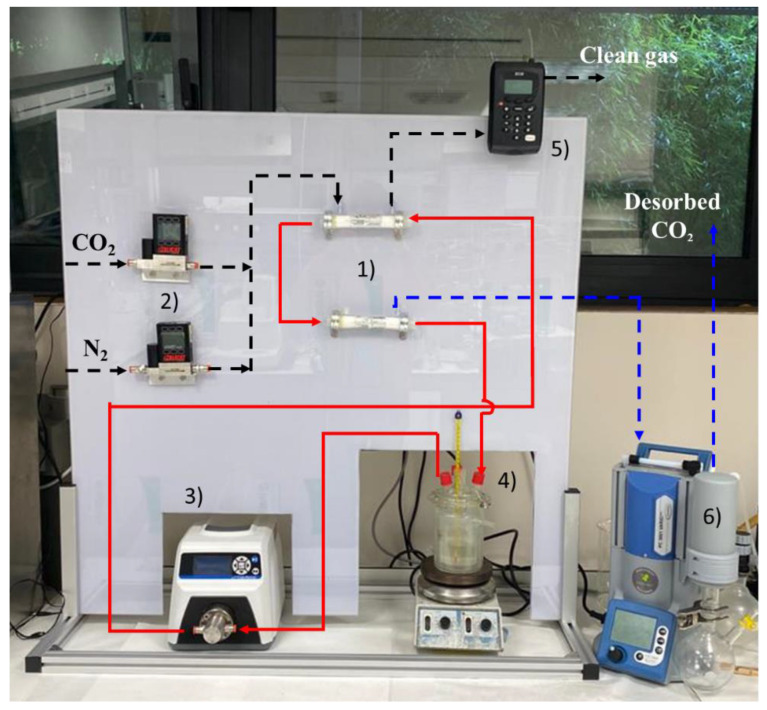
Experimental set-up of continuous absorption–desorption process using MVR technology.

**Figure 3 membranes-12-00785-f003:**
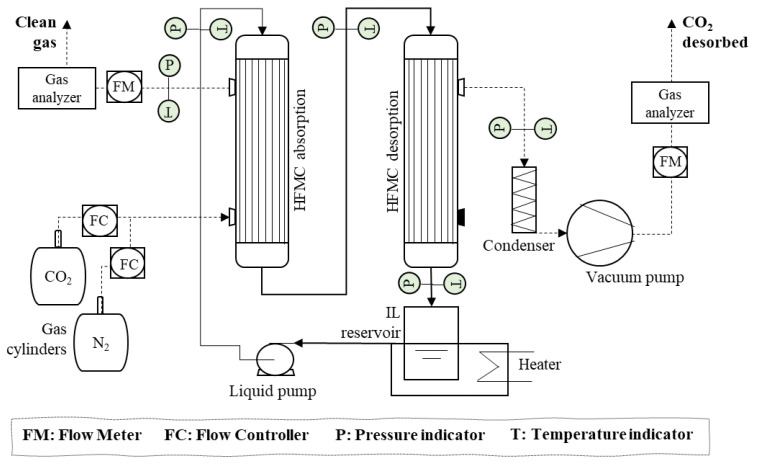
Experimental setup of the CO_2_ absorption–desorption process with one absorption HFMC and one desorption HFMC for MVR. Gas flow (dashed lines), Liquid flow (solid lines).

**Figure 4 membranes-12-00785-f004:**
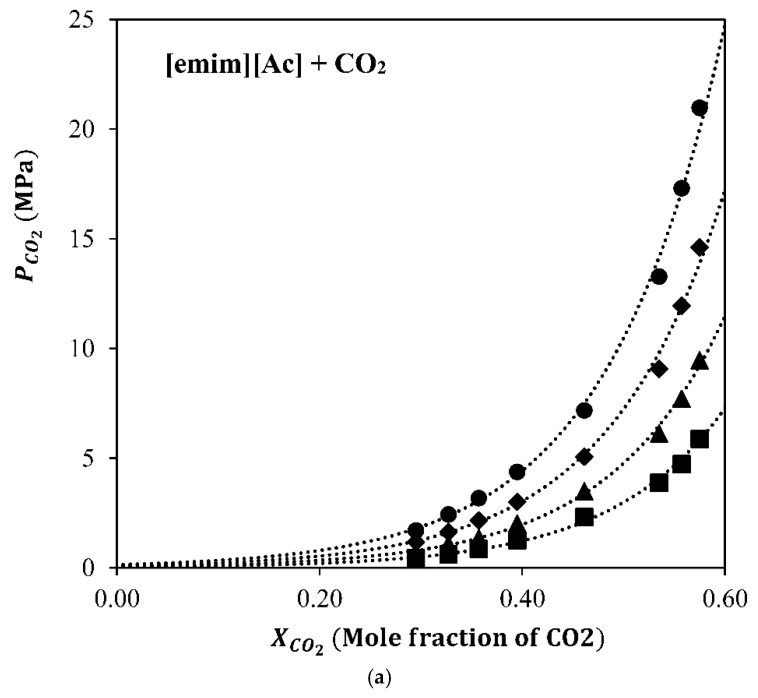
P−X diagram of CO_2_ solubilities of the ionic liquid + CO_2_ system ((**a**) [emim][Ac], (**b**) [emim][MS]). The symbols represent the temperatures, respectively; (●) 303.15 K, (♦) 323.15 K, (▲) 343.15 K, and (■) 363.15 K. Adapted with permission from Yim et al. [28]. Copyright (2018) American Chemical Society.

**Figure 5 membranes-12-00785-f005:**
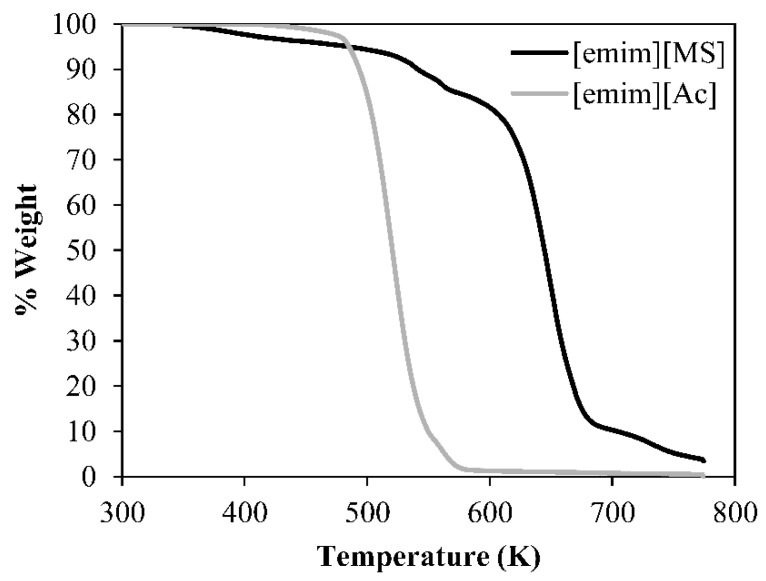
TGA analysis for the [emim][MS] and [emim][Ac] ionic liquids.

**Figure 6 membranes-12-00785-f006:**
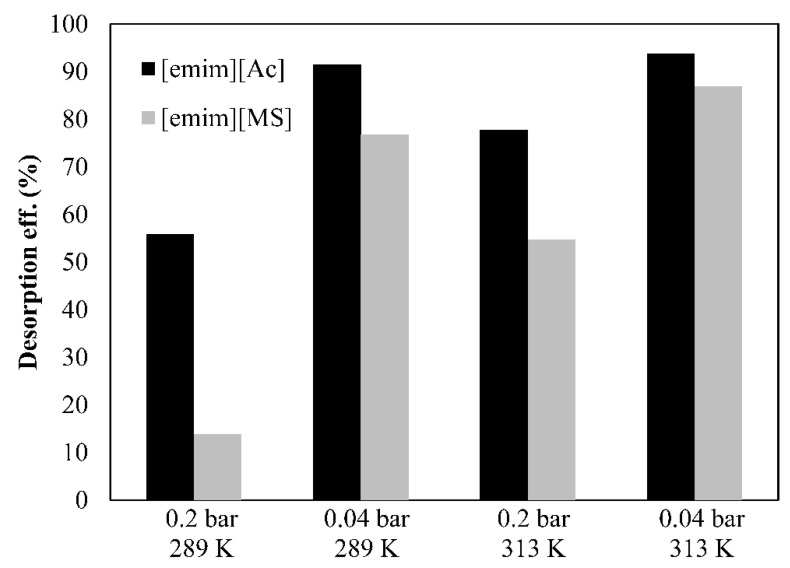
Desorption efficiency by using 2 ILs at different vacuum pressures and temperatures. Commercial HFMC operational conditions: liquid flow rate 60 mL·min^−1^.

**Figure 7 membranes-12-00785-f007:**
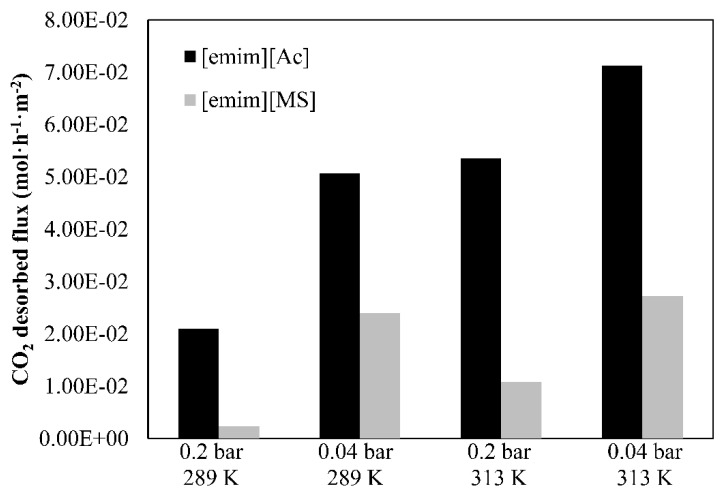
CO_2_ desorbed flux by using 2 ILs at different vacuum pressures and temperatures. Commercial HFMC operational conditions: liquid flow rate 60 mL·min^−1^.

**Figure 8 membranes-12-00785-f008:**
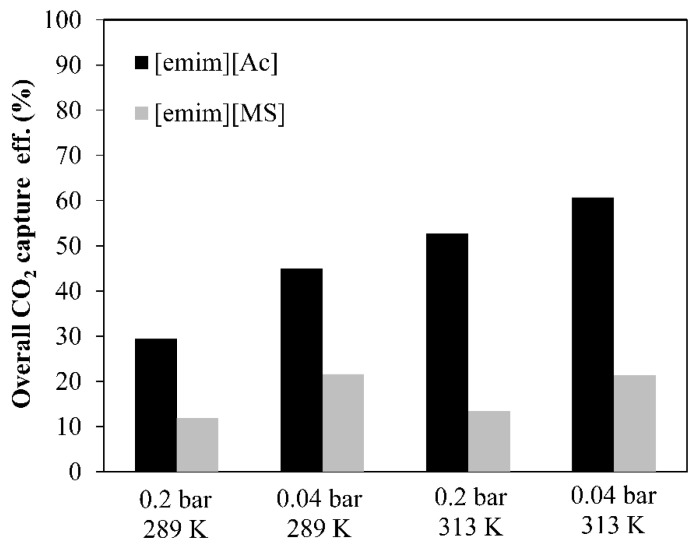
Overall CO_2_ capture performance by using 2 ILs at different vacuum pressures and temperatures. Commercial HFMC operational conditions: liquid flow rate 60 mL·min^−1^.

**Figure 9 membranes-12-00785-f009:**
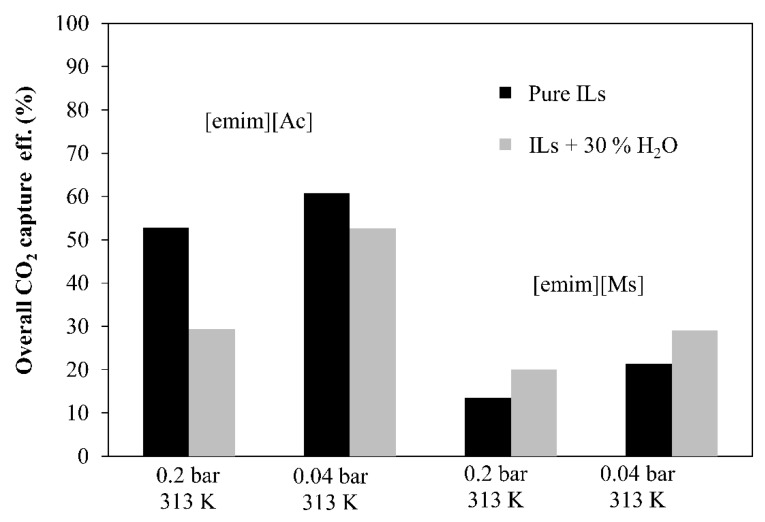
Overall CO_2_ capture efficiency at different desorption vacuum pressures using both pure ILs and aqueous ILs. Commercial HFMC operational conditions: temperature 313 K, liquid flow rate 60 mL·min^−1^.

**Table 1 membranes-12-00785-t001:** Abbreviation, molecular formula, and chemical structure of the two ILs studied.

Abbreviation	Molecular Formula	Chemical Structure
1-ethyl-3-methylimidazolium acetate [emim][Ac]	C_7_H_14_N_2_O_4_S	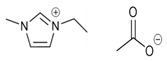
1-ethyl-3-methylimidazolium methyl sulfate [emim][MS]	C_7_H_14_N_2_O_4_S	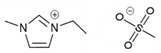

**Table 2 membranes-12-00785-t002:** Hollow fiber membrane contactor (HFMC) characteristics (1 × 5.5 MiniModule TM).

Parameter	Value
Membrane Material	Polypropylene
Module configuration	Parallel
Module i.d., d_cont_ (m)	25 × 10^−3^
Fiber outside diameter, d_o_ (m)	3 × 10^−4^
Fiber inside diameter, d_i_ (m)	22 × 10^−5^
Fiber length, L (m)	0.115
Number of fibers, n	2300
Effective inner membrane area, A (m^2^)	0.180
Membrane thickness, δ (m)	4 × 10^−5^
Membrane pore diameter, dp (m)	4 × 10^−8^
Porosity, ς (%)	40
Packing factor, φ	0.390
Tortuosity, τ	2.500

**Table 3 membranes-12-00785-t003:** Operating conditions of the absorption–desorption process based on the non-dispersive gas–liquid HFMC contactors, laboratory scale.

Parameter/Property	Value	Unit
Volume, V	250	mL
Temperature, T	289–310	K
Feed Gas flow rate, F_g_	60	mL·min^−1^
Liquid flow rate, F_l_	60	mL·min^−1^
Feed gas pressure, P_g,in_	1.03	bar
Liquid pressure, P_l,in_	1.31	bar
Vacuum pressure, P_v_	0.04–0.50	bar

**Table 4 membranes-12-00785-t004:** Viscosity of [emim][Ac] and [emim][MS] and contact angle measured between the membrane and the ILs.

Property	[emim][Ac]	[emim][MS]
Viscosity, pure ILs, cP	138	48
Viscosity, ILs + 30% H_2_O, cP	12.1	5.8
Measured contact angle (°)	114.5	110.5

## Data Availability

Not applicable.

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
