# Peer review of "Chemical and Physical Ionic Liquids in CO2 Capture System Using Membrane Vacuum Regeneration"

_membranes, 2022, doi:10.3390/membranes12080785_

Round 1

Reviewer 1 Report

The present work introduced a novel CO2 capture technology strategy based on non-dispersive CO2 absorption and membrane vacuum regeneration (MVR) technology. Two imidazolium ionic liquids (ILs) to study their nature behaviour on continuous absorption-desorption performance. The chemical IL [emim][Ac] shows better CO2 desorption performance and higher CO2 desorbed flux than physical IL [emim][MS] at the operation conditions studied. Herein, this work is recommended to be published after revision. The suggestions are listed as below.

Suggestions:

1.     May the author justify the selection of polypropylene as the polymeric hollow fiber membrane material over other thermal resistant materials (e.g., polyamide and polyvinylidene fluoride (PVDF))?

2.     May the author justify the choice of [emim]-based ILs as physical and chemical ILs in this study?

3.     May the author provide a schematic showing the absorption-desorption process for both ILs?

4.     How can the author ensure the commercial viability of this membrane contactor during scale-up implementation as the studied temperature in overall CO2 absorption is only in the range of 289-313 K.

5.     How can the authors confirm the industrial feasibility of this work since the recovery of chemical IL required large amount of energy? Also, is the CO2 capture efficiency of 61% sufficient to evidence the industrial implementation?

6.     Adding water into ILs could promote the dissociation of ILs which changes the structure of ILs, eventually limits the efficiency of ILs in CO2 absorption. May the authors explain the mechanism whereby CO2 absorption of physical ILs is not affected by water.

7.     Figure 9: [emim][Ac] is not properly labeled in the figure and this may lead to the misunderstanding of the reader. Kindly revise.

8.     To broaden the perspective of this article, please read and include Membranes, 2022, 12, 539.

Author Response

  1. Polypropylene hollow fibers are recommended for temperatures 289-313 K The inlet gas has to be feed in this range of temperature. The energy recovery in the plant may use the content of energy onf the effluent.
  2. Based on previous studies, which are cited in the text emim based IL are a compromise among different variables of interest.
  3. Figures 3 and 4 show the experimental system and the technical application of the absorption-desorption. In the paper it has not been studied the molecular mechanism.
  4. The control of the temperature is a restriction of the system to be considered in the implementation
  5. In a previous paper it has been calculated the energy required in this process, which is musch lower than other absorption alternatives. In the  case of the demand of higher efficiencies more membrane area would be necessary. But it is possible
  6. The paper des not study mechanisms, the scope is related to the process.
  7. Figure 9 has been revised
  8. We have included in the paper the reference with the number 36

Reviewer 2 Report

Review of membranes-1852539

This is an interesting manuscript about two different ionic liquids (ILs), i.e. (1) chemical IL [emim][Ac], (2) physical IL [emim][MS] for the application of CO2 capture system using membrane vacuum regeneration (MVR) technology. Detailed characterizations and experiments have been performed for this study. However, there are some issues that must be clarified, as follows:

  1. Figure 2, part number 4; and Figure 3 (IL reservoir): What is the volume of each IL employed in this study? Please mention it in the manuscript.
  2. Related to the ILs, and their future for the CCUS (carbon capture, utilization, and sequestration), where will the CO2 be stored? In the ionic liquid, or in other material(s)? Please mention this issue in the manuscript.
  3. Related to the volume of ILs, and its future for the global urgency of CCUS to slow down the global warming, definitely there will be a high demand for a high volume of ILs if this MVR technology is successful. Please write in the manuscript about the predicted volume of the ILs that will be demanded, to capture how much CO2 per year.
  4. Based on the aforementioned predicted volume of CO2 and ILs, please add some simple economical analysis (it does not have to be profitable) about the price of [emim][Ac], [emim][MS], and the aqueous [emim][MS] against the price of CO2 captured. This economic analysis is crucial in order to demonstrate to the public about the prospect of this technology.
  5. Figure 1, 2, and 3: How about the swelling of polypropylene hollow fiber membranes when in contact with each IL?
  6. Line 88: CO2 --> with subscripted 2
  7. Line 106, 106, 109, 113: --> please write trademarked brands with the proper ™ symbol
  8. Line 108 and 112: Please delete the long name of the ionic liquids that have been mentioned previously in lines 92 and 93. If you insist, please put the long name inside Table 1.
  9. Line 137, 138, 144: --> please write trademarked brands with the proper ™ symbol
  10. Line 154: Alicat Scientific --> with uppercase S
  11. Line 154, 157, 160, 162, 166: --> please write trademarked brands with the proper ™ symbol
  12. Line 158: What do you mean with “ml/rev”? Please clarify. In addition, please write it as mL rev-1, (with uppercase L and superscripted -1), in order to be consistent with the writing of the units in this manuscript, especially in Tables 2 and 3.
  13. Figure 4: Please label the figures and Figure 4a and 4b. Please make the axis in consistent style for Figure 4a, with two numbers behind the decimal point, like that of Figure 4b.
  14. Figure 4a and 4b: Please write the label of x axis with CO2 --> with subscripted 2.  
  15. Line 323: …by Equation 3… in Section 2.4 --> with uppercase E, and add the word “Section”
  16. Line 361: …by Equation 3… --> with uppercase E
  17. Figure 9: Delete the legend “[emim][MS]”. Rewrite the legend, please. Change “pure ILs” with “[emim][Ac]”, and change “IL + 30% H2O” with “[emim][MS] + 30% H2O”. In this way, the figure is easier to be digested and not generating confusion like its previous version.
  18. Figure 9 and Table 4: Please add the data of the viscosity of [emim][MS] + 30% H2O
  19. Reference 10: Please delete “(Basel)”
  20. Reference 26: CO2 --> with subscripted 2
  21. Reference 28: Please write the name of the ionic liquids correctly! SO4 with subscripted 4.
  22. Reference 31: Please write the name of the ionic liquids correctly! SO4 with subscripted 4, NTf2 with subscripted 2, and N(CN)2 with subscripted 2.
  23. Reference 31: The DOI number is wrong. The correct DOI number is: 10.1021/jp804319a --> without “32” at the back of it.
  24. DOI numbers must be written in a consistent style, not some with “https://doi.org”  or with “doi:10.xxx”. Please write them consistent with one style only.

Author Response

In the attached document the comments have been answered

Round 2

Reviewer 2 Report

Review of membranes-1852539-v2

The authors have addressed the issues well, the manuscript can be accepted now.